# Feline Facial Spindle Cell Tumors in 29 Cats: Histomorphological and Immunohistochemical Characterization

**DOI:** 10.3390/ani14071103

**Published:** 2024-04-04

**Authors:** Sara Soto, Sohvi Blatter, Stefan Hobi, Marco Steck, Julia Lechmann, Silvia Rüfenacht, Nicolas Kühn, Maja Ruetten, Nataliia Ignatenko, Christiane Krudewig

**Affiliations:** 1Institute of Animal Pathology, Vetsuisse Faculty, University of Bern, 3012 Bern, Switzerland; 2Department of Veterinary Clinical Sciences, Jockey Club College of Veterinary Medicine and Life Sciences, City University of Hong Kong, Kowloon Tong, Hong Kong SAR, China; 3Institute of Virology, Vetsuisse Faculty, University of Zurich, 8057 Zurich, Switzerland; 4DermaVet, 5036 Oberentfelden, Switzerland; 5Kühn Pathologie AG, 6331 Hünenberg, Switzerland; kuehn@kuehnpathologie.ch; 6PathoVet CH, 8317 Tagelswangen, Switzerland; maja.ruetten@pathovet.ch; 7Centre for Clinical Veterinary Medicine, University of Munich, 80539 Munich, Germany; natali.ignatenko@gmail.com; 8Institute of Veterinary Pathology, Vetsuisse Faculty, University of Zurich, 8057 Zurich, Switzerland

**Keywords:** facial, feline, immunohistochemistry, oncology, peripheral nerve sheath tumor, sarcoid, skin, soft tissue sarcoma, soft tissue tumor, Sox10

## Abstract

**Simple Summary:**

This study aims to characterize soft tissue tumors/sarcomas (STSs) occurring on the faces of cats using histomorphology, immunohistochemistry, and molecular (PCR) techniques. By examining 34 tumors from 29 cats, we found that many were peripheral nerve sheath tumors (PNSTs), identified by specific protein markers. Interestingly, only a few tumors thought to be sarcoids were confirmed by testing for BPV14. This highlights the need for careful diagnostic methods to correctly identify these tumors. Some tumors appeared as unusual lesions on the face, not typical masses, pointing out an important clue for clinicians. The study also introduces a new observation in tumor cells’ pattern and underscores the value of the Sox10 marker in diagnosing PNSTs in cats, offering new insights for veterinary pathologists.

**Abstract:**

Soft tissue tumors/sarcomas (STSs) in felines, encompassing a variety of mesenchymal tumors with similar histomorphological features, present diagnostic challenges due to their diverse cellular origins and the overlap with other tumor types such as feline sarcoid. This study aimed to delineate the clinical, histomorphological, and immunohistochemical characteristics of 34 feline facial spindle cell tumors affecting 29 cats, including testing for bovine papillomavirus type 14 (BPV14), the virus causing feline sarcoids. Only five out of 12 tumors previously diagnosed as feline sarcoids based on histomorphology were confirmed by PCR for BPV14, underscoring the importance of comprehensive diagnostic approaches to accurately distinguish between STSs and feline sarcoids. This study shows that most facial spindle cell tumors were compatible with peripheral nerve sheath tumors (PNSTs) based on positive immunohistochemical staining for Sox10 and other immunohistochemical markers such as GFAP, NSE, and S100. Some of these tumors displayed as multiple independent masses on the face or as erosive and ulcerative lesions without obvious mass formation, an atypical presentation and an important highlight for general practitioners, dermatologists, and oncologists. This study also describes periadnexal whorling of neoplastic cells as a novel histomorphologic finding in feline facial PNSTs and emphasizes Sox10 as a useful complementary immunohistochemical marker for the diagnosis of facial PNST in cats, providing valuable insights for veterinary pathologists.

## 1. Introduction

Soft tissue tumors, or soft tissue sarcomas (STSs), are general terms used for mesenchymal tumors of different cellular origins displaying similar histomorphological features. Furthermore, a similar clinical behavior of the tumors in this group has been observed [1,2,3]. Terminology is not clearly defined in feline STSs. However, included in this group are peripheral nerve sheath tumors (also known as nerve sheath tumors), fibrosarcoma, myxosarcoma, leiomyosarcoma, liposarcoma, rhabdomyosarcoma, and unspecified spindle cell tumors/sarcomas arising in the dermis or subcutis, while the inclusion of perivascular wall tumors and feline injection site sarcoma is controversially discussed [2]. Despite its low specificity, the term STS is still favored in veterinary medicine based on the difficulty, mainly economical, of achieving a more precise diagnosis, as expensive immunohistochemical panels would be required for a more concrete diagnosis [1,2,3]. However, further studies with defined criteria and subtyping of STSs are needed to prove the hypothesis of a common behavior and to explore cell-type-specific development and treatments. In this sense, an accurate tumor diagnosis should be aimed for, as this is still one of the main predictors of outcome in medical oncology [4,5]. In addition, this is also of special interest for comparative pathology, as, in contrast to small domestic animals such as cats, this tumor is rarely diagnosed in human medicine [6].

Feline sarcoid is a tumor caused by bovine papillomavirus type 14 (BPV14) that occurs mainly on the faces of cats [7] and presents local infiltrative growth but does not metastasize [8]. This tumor type is not included in the group of feline STSs but displays important overlapping of histomorphological features with low-grade STS, complicating the diagnosis of spindle cell tumors in this location even further [2]. Differentiation is important as feline sarcoids can differ from STSs concerning prognosis, and their viral etiology has clinical significance [9]. The detection of BPV14 is used for confirmation of viral origin and the diagnosis of feline sarcoid [9]. Nevertheless, techniques to detect BPV14 are cost-intensive, not always available, and therefore rarely applied in clinical settings. This study aims to characterize 34 facial STSs affecting 29 cats, including their clinical features and progression of lesions over time, using histomorphology, immunohistochemistry techniques, and BPV14 PCR analysis.

## 2. Material and Methods

### 2.1. Sampling and HE-Findings

Thirty-four cutaneous tumors affecting the faces of 29 cats, which were diagnosed as “soft tissue tumor”, “soft tissue sarcoma”, “spindle cell tumor”, “peripheral nerve sheath tumor”, or “feline sarcoid”, have been selected for this study. All tumors have been sampled as incisional or excisional biopsies between 2013 and 2023. They have been selected from the databases of the Veterinary Pathology Institutes at the Veterinary Faculties in Bern and Zürich (Switzerland), the Veterinary Medical Centre (VMC) of the Jockey Club College of Veterinary Medicine and Life Sciences of the City University of Hong Kong (Kowloon Tong, Hong Kong SAR, China), the private pathology laboratories Kühn Pathologie and PathoVet (Switzerland), and a private veterinary clinic in Kyiv (Ukraine). Clinical information on the animals was obtained from the submission forms accompanying the biopsy samples. All tumors had been fixed in 10% buffered formalin, trimmed, embedded in paraffin, sectioned at 2–3 µm thickness, and stained with hematoxylin and eosin (H&E). The H&E slides of the tumors were collected, and the following histopathological features were reassessed by a board-certified pathologist: 1. presence or absence of prominent epidermal hyperplasia forming long rete ridges and intimal association of the tumor with the epidermis; 2. presence of nerve sheath growth patterns, including Antoni A and Antoni B regions, Verocay bodies, and whorls [10]; 3. histopathological grading according to the grading system for cutaneous and subcutaneous soft tissue tumors/sarcomas (STS) proposed by Dobromylskyj et al. 2021 [11]. Microscopic images were performed using an Olympus^®^ BX51 microscope (Olympus Schweiz AG, Wallisellen, Switzerland) and an Olympus^®^ DP27 camera (Olympus Schweiz AG, Wallisellen, Switzerland). No animal experiments have been performed for this study.

### 2.2. Immunohistochemistry

Immunohistochemistry was performed using the markers vimentin, SRY-related HMG-Box gene 10 (Sox10), S100, glial fibrillary acidic protein (GFAP), neuron-specific enolase (NSE), periaxin, smooth muscle actin (SMA), p63, MelanA, and melanoma-associated antigen (PNL2). Detailed information on the antibodies used and their respective immunohistochemical protocols is described in Table 1. For the markers vimentin, Sox10, S100, GFAP, NSE, SMA, p63, Melan A, and PNL2, 2–3 µm formalin-fixed and paraffin-embedded (FFPE) sections using positively charged slides were dried for 35 min at 60 °C and subsequently dewaxed, pretreated for antigenic retrieval, and stained on a Bond-III immunostainer (Leica Biosystems^®^, Biosystems Switzerland AG, Muttenz, Switzerland). After dewaxing (Bond Dewax solution; Leica Biosystems^®^), a pretreatment for antigenic retrieval, detailed in Table 1, was performed. To reduce the non-specific binding of primary antibodies, a protein block solution was applied for 10 min at room temperature. This temperature was used for all the following steps. Afterwards, the slides were incubated with the primary antibody for 15 min. For the antibodies vimentin, Sox10, S100, GFAP, NSE, SMA, and p63, the following steps were performed using reagents from the Bond Polymer Refine Detection Kit (Leica Biosystems^®^): Endogenous peroxidase was blocked for 5 min, then a secondary antibody was applied (8 min), followed by a peroxidase-labeled polymer (8 min). These reagents were supplemented with 2% dog serum to block non-specific binding (LabForce^®^, LabForce AG, Muttenz, Switzerland). Finally, slides were developed in 3,3′-diaminobenzidine/H_2_O_2_ (10 min). For the antibodies MelanA and PNL2, after incubation with the primary antibody, the following steps were performed using reagents of the Bond Polymer Refine Red Detection Kit (Leica Biosystems^®^): A secondary antibody was applied (20 min), followed by a polymer AP (30 min). These reagents were supplemented with 2% dog serum to block non-specific binding (LabForce^®^)All slides were then counterstained with hematoxylin and mounted.

For periaxin immunohistochemistry, 2–3 µm thick FFPE sections using positively charged slides were dried at 37 °C overnight. After deparaffinization, a heat and pressure pretreatment for antigenic retrieval was performed using EDTA buffer (pH 9) in a pressure cooker at 98 °C for 20 min. The slides were then rinsed with distilled water and put into TBS-Tween buffer (Dako^®^ 3006, Fisher Scientific AG, Reinach, Switzerland). The subsequent steps were performed with a Dako^®^Autostainer at room temperature following this protocol: incubation with the periaxin antibody for 60 min, rinsing with TBS-Tween buffer (Dako^®^ 3006), peroxidase blocking buffer (Dako^®^ S2023) for 10 min, rinsing with TBS-Tween buffer (Dako^®^ 3006), Envision+System HPR Rabbit (Dako^®^ K4003) for 30 min, rinsing with TBS-Tween buffer (Dako^®^ 3006), and developing of the slides with DAB (Dako^®^ K3468) for 10 min. Afterwards, the slides were rinsed with distilled water, counterstained with hematoxylin, and mounted. 

Known feline positive control tissue was stained in parallel with each series of slides (adrenal gland for S100 and NSE; spinal cord for GFAP; known melanoma for MelanA, PNL2, and Sox10; mammary gland for p63; subcutis/intestine for vimentin and SMA; known PNST for periaxin). Negative controls, in which the primary antibody was replaced with wash buffer, were also utilized in all cases.

The immunohistochemistry was semi-quantitatively assessed by the same board-certified pathologist. A tumor was considered positive for an antibody if at least approximately 10% of the neoplastic cells were stained.

Microscopic images were performed as described in the section “Sampling and HE-Findings”.

### 2.3. PCR for Bovine Papillomavirus 14 (BPV14)

DNA was extracted from paraffin-embedded tumor tissue using the QIAamp DNA Mini Kit (Qiagen, Hombrechtikon, Switzerland) according to the manufacturer’s instructions. For tumor 26 (cat 21), previously reported primer sets jmpSA-for (5′-GGAACAAACCTCACAATCAC-3′) and jmpSA-rev (5′-CCAGTTCTCTAATACTGAGG-3′) amplifying a 195 bp product in the L1 region (6612 to 6806) of the BPV14 genome [12], as well as additional primers amplifying a 549 bp product in the L1 region (5771 to 6319) of the BPV14 genome BPV14for (5′-TGG TAA AGA GGT GCC CAA AG-3′) and BPV14rev (5′-GCT TCC TCA GCC ATTTTG AG-3′) were used [13]. PCR was performed with a reaction mix of 8 µL water, 2 µL of each forward and reverse primer (10 µM each), 1 µL extracted DNA as template, and 12 µL REDTaq ReadyMIX (SIGMA-ALDRICH, Buchs, Switzerland) in a total volume of 25 µL. For the cats with other tumors, which were analyzed later in time, the newly designed primer sets BPV14for new (5′-GCA GCA AAA ACT GCC TTT TC-3′) and BPV14rev new (5′-TAT AAT CCC ACG CAA CGT GA-3′) amplifying a 264 bp product in the upstream regulatory region (7291 to 7554) were used. PCR was performed with a reaction mix of 14 µL nuclease-free water (VWR), 500 nM of each forward and reverse primer, 5 µL extracted DNA, 2.5 µL 10× PCR Buffer (HotStarTaq DNA Polymerase, Qiagen^®^, Qiagen AG, Hombrechtikon, Switzerland), 0.5 µL dNTP (10 mM, Thermo Scientific^®^, Thermo Fisher Scientific, Basel, Switzerland and 2.5U HotStarTaq DNA Pol (5 U/µL) (Qiagen^®^) in a total volume of 25 µL.

Detection of the GAPDH gene, using the primers catGAPDHfor (5′-TCA TCA TCT CTG CCC CTT CT-3′) and catGAPDHrev (5′-GTG AGC TTC CCA TTC AGC TC-3′) amplifying a 330 bp product served as an extraction control.

The cycling program for the PCR assays started with a denaturation step of 3 min at 94 °C, followed by 40 cycles of 30 s at 94 °C, 30 s at 55 °C, and 30 s at 72 °C. PCR products were visualized in a 1.5% agarose gel (standard agarose-type LE, Bioconcept^®^, Bioconcept AG, Allschwil, Switzerland). Bands of the expected size were excised, extracted (QIAquick gel extraction kit, Qiagen^®^, Hombrechtikon, Switzerland), and purified using the QIAquick PCR purification kit (Qiagen^®^) according to the manufacturer’s instructions. Nucleotide sequences were determined (Microsynth, Balgach, Switzerland) and compared with the published reference sequences of BPV14 using the NCBI Basic Local Alignment Search Tool (“BLAST”) (http://www.ncbi.nlm.nih.gov/blast/Blast.cgi, accessed on 1 November 2023). The sequencing results showed 99% and 100% identity to the published BPV14 sequences (Genbank accession #KP276343).

### 2.4. Follow-Up

Information about treatment, relapse of the tumors, and mortality, including cause and date of death, was requested per mail and/or phone from the veterinarians that had submitted the biopsies of the cats.

## 3. Results

### 3.1. Clinical Findings

The clinical findings are detailed in Table 2. Of the twenty-nine cats included in the study, there were sixteen European shorthair (ESH) (55.2%), six domestic shorthair (DSH) (20.7%), three Maine Coon (10.3%), two British shorthair (BSH) (6.9%), one Canadian Sphynx (3.4%), and one mixed breed cat (3.4%). From these, nineteen were male (65.5%) and nine were female (31%), while the sex was unknown in one case. The age was collected for all except three cats, varying from 9 months to 16 years, with a mean of 9.2 years and a median of 11 years.

Twenty-four cats (82.8%) presented with a single tumor, while four cats (13.8%) presented with two or three clinically independent tumors at different locations on the face (Figure 1). One cat (3.4%) displayed poorly demarcated lesions (Figure 1a), leading to a final score of 34 tumors. Of these 34 tumors, 28 (82.4%) presented as a mass, one (2.9%) as a plaque, four (11.8%) as erythematous to crusty erosion, ulceration, or as plaque further progressing to a mass, and one (2.9%) as poorly demarcated crusts, erosions, and ulcerations. Seventeen tumors (50.0%) were located on the lips or adjacent skin (eight (23.5%) at the upper lip, one (2.9%) at the inferior lip, eight (23.5%) not specified); nine (26.5%) at the nose (three (8.8%) on the nasal bridge, one (2.9%) on the nasal planum, one (2.9%) on the nasal planum and the nasal bridge, two (5.9%) at the right nares, one (2.9%) at the left nares, one (2.9%) not specified); one (2.9%) at the nasal planum and upper lip; three (8.8%) at the cheeks; two (5.9%) at the forehead; one (2.9%) at the left medial canthus extending in the nearby upper and inferior eyelids; and one (2.9%), the tumor presenting as poorly demarcated crusts, erosions and ulcerations, from the right medial canthus over the nasal bridge and nasal planum to the right and left nares. Clinical differential diagnoses indicated by the primary clinicians were viral (Calicivirus, Herpesvirus, Papillomavirus/feline sarcoid), fungal (cryptococcosis), allergic (eosinophilic granuloma complex), and neoplastic (squamous cell carcinoma, feline sarcoid, not further specified).

### 3.2. Pathological Findings

The pathological findings are detailed in Table 3. All 34 tumors consisted of a moderately to poorly demarcated, unencapsulated, infiltrative neoplastic proliferation of long to plump spindle cells arranged in disorganized bundles in scant to moderate amounts of stroma, rarely mimicking an Antoni B pattern. No Antoni A areas or Verocay bodies were observed. In 11 tumors (32.4%), the neoplastic cells were partly arranged in whorls, especially around the skin adnexal units. This periadnexal whorling, together with prominent epidermal hyperplasia, were the main findings in cat 1 with the poorly demarcated skin lesions (Figure 2). Thirty-two (94.1%) tumors in the study presented low to moderate anisocytosis and anisokaryosis, a low to moderate number of mitoses, no to mild necrosis, and no to mild inflammation, and were graded as grade 1 (27 tumors, 79.4%) or grade 2 (five tumors, 14.7%) following the grading system proposed by Dobromylskyj et al. [11] (Figure 3). Ten of the grade 1 tumors (37%) and two of the grade 2 tumors (40%) were histopathologically diagnosed or presumed to be feline sarcoids based on the intimal association of the tumor with the overlying epidermis and the prominent epidermal hyperplasia with the formation of long rete ridges. The other grade 1 and grade 2 tumors were diagnosed as STSs. Two tumors (5.9%) in the study presented moderate to high anisocytosis and anisokaryosis, a high number of mitoses (more than 20 mitoses in a tumor area of 2.37 mm^2^), and areas of tumor necrosis and were diagnosed as grade 3 STS with amelanotic melanoma as a differential diagnosis (Figure 3).

In the immunohistochemical analysis, all antibody epitopes displayed cytoplasmic expression except the epitopes for Sox10 and p63, which are expressed in the nucleus. All the assessed tumors were positive for vimentin and negative for Melan-A. PNL-2 was negative in the 25 tumors that were tested. Twenty-five tumors (73.5%) were positive for Sox10 (Figure 2 and Figure 3). From these, one (4%) was also positive for S100, GFAP, NSE, and periaxin (Figure 3), four (16%, one of them grade 3) also for S100, GFAP, and NSE, 10 (40%) also for S100 and GFAP, one (4%) also for S100 and NSE, and eight (32%, one of them grade 3) also for S100. From the tumors showing positivity for Sox10 and at least one of the other previously described antibodies (S100, GFAP, NSE, and periaxin), four were also positive for SMA, two also for p63, and one for both SMA and p63 (Figure 2). One Sox10-positive tumor (4%) only showed additional positivity for SMA. Altogether, these 25 Sox10-positive tumors were interpreted as PNSTs. From the nine tumors negative for Sox10, one (11.1%) was positive for S100 and SMA, and one (11.1%) for NSE. Both were interpreted as PNSTs too, making a total of 27 PNSTs (79.4%) in our study. Another tumor (11.1%) was positive for SMA and interpreted as a leiomyosarcoma, while the other six (66.7%) tumors were negative for all tested antibodies except vimentin. 

In the four cats with multiple tumors, all tumors were positive for Sox10. Only in cat 3, all masses showed the same immunohistochemical profile, with positivity for Sox10 and S100, while in the other cats, the tumors exhibited different immunohistochemical features: cat 10 had one cheek mass displaying positivity for Sox10, S100, and GFAP, while the other one stained positive for Sox10 and SMA; the plaque of the right nares from cat 20 was positive for Sox10, S100, and GFAP, while the mass of the lip was only positive for Sox10 and S100; the plaque and the mass of the left canthus and left lip from cat 12 were positive for Sox10 and S100, while the mass of the cheek was positive for Sox10, S100, GFAP, and SMA. 

The 11 tumors with neoplastic whorling around the adnexal units were all positive for Sox10. Similar to what was described for the immunohistochemistry in the cats with multiple tumors, only in two cats did both tumors present whorling around the adnexal units, while in the other two, it was observed only in one of the tumors.

### 3.3. PCR for Bovine Papillomavirus 14 (BPV14)

In five tumors (14.7%), which based on histopathological features were all compatible with feline sarcoids and negative for all tested antibodies except vimentin, the primer sets for feline sarcoid virus BPV14 amplified bands of the expected size. The sequencing results showed 99% and 100% identity to the published BPV14 sequences (Genbank accession #KP276343). In three tumors that histomorphologically were compatible with STS or amelanotic melanoma, cat GAPDH was not detectable, and PCR results could not be interpreted. The other tumors tested negative, including those in the cats presenting more than one tumor.

### 3.4. Follow-Up

Complete or partial follow-up information was obtained from 17 cats (58.6%). A total of 17.6% of these cats, including two of the three cats with feline sarcoid and the cat with the tumor negative for BPV4 and negative for all antibodies except vimentin, were alive at the time of the study, reflecting a time frame from 2.3 to 7.5 years (832 to 2732 days). One cat with feline sarcoid died, but no further information about the date or cause of death was available. The other 13 cats that died during the time frame of the study had tumors interpreted as PNST (76.5%), 11 of them positive for Sox10. Eight out of these 13 cats (61.5%) were euthanized due to poor life quality caused by the tumor, with a survival time from diagnosis (based on biopsies) to date of death varying from 1 to 760 days (mean of 190 days and median of 115 days); two (15.4%) due to diseases independent of the facial tumor (cardiomyopathy and abdominal lymphoma), while for the other three animals (23%) the cause of death was unknown. Relapse of the tumor was described in four of the 17 cats with follow-up (23.5%), all interpreted as PNST. Apart from the surgical resection, only one out of the 17 cats (5.9%) was treated with chemotherapy and radiotherapy (protocols unknown). To exclude metastatic disease, five cats (29.4%) received thoracic radiography, but only in one of them (20%) suspected distant metastases were found in the liver.

## 4. Discussion

The purpose of this study was to assess and describe the clinical, histomorphological, and immunohistochemical features of 34 feline facial spindle cell tumors, including their BPV14 status (virus causing feline sarcoid), since a detailed description of feline facial spindle cell tumors is lacking. Based on PCR results, out of 12 feline sarcoids previously diagnosed by histomorphology alone, only five (41.7%) could be confirmed using PCR for BPV14, known to be currently the most sensitive technique to confirm the diagnosis of a feline sarcoid [9]. In addition, these five confirmed feline sarcoids were immunohistochemically negative for all tested antibodies except vimentin, further supporting their fibroblastic origin [7,14]. Although the BPV14 PCR was inconclusive in three tumors, they were less likely to be feline sarcoids based on immunohistochemistry and/or old age. The remaining tumors were all negative for BPV14. 

The five cats with confirmed feline sarcoid were young (up to 2.5 years old; excluding these cases, the mean and median age of the cats in our study ascended from 9.2 years (mean) and 11 years (median) up to 11 years (mean) and 12 years (median)). They also showed masses on the lips (3 tumors) and nasal planum/nares (2 tumors), which agrees with previous studies [7,8,9,14]. This may reflect the territorial marking behavior presented by young outdoor cats getting into territorial fights [8]. As expected for this tumor, of the three cats with confirmed feline sarcoid and available follow-up, two were still alive at the time of the study (>2 years), while the third one died due to unknown causes [7,8,9,14].

Interestingly, one of the investigated tumors, negative for BPV14 and all immunohistochemical markers except vimentin (cat 26), displayed typical histopathological and clinical features, including location and age, for a feline sarcoid. Nevertheless, this cat was confirmed to be indoors only (in Hong Kong) with no access to farm animals. Similarly, the owner of this cat did not have any contact with bovines and did not feed the cat raw meat. Based on this, and as the PCR extraction control (cat GAPDH) for this case was positive, we believed the PCR results to be true. This tumor was then interpreted as a grade 1 feline sarcoid-like fibrosarcoma. It was surgically resected, and the affected area as well as the cat were clinically unremarkable 2.3 years after the tumor diagnosis. Similar suspected feline sarcoids negatively tested for papillomavirus with PCR have been previously published [14,15,16], but in these studies, the accessibility of the cats to farm animals and their age were not reported. These findings emphasize the point that feline sarcoid-like tumors with no evidence of BPV14 infection can occur in young cats. A possible papillomavirus infection with a virus type that may be beyond the detection capacity of the applied PCRs cannot be excluded. 

The other six tumors in our study that had been wrongly diagnosed as feline sarcoids based on histomorphology only were all positive for Sox10 and S100, and part of them also presented positivity for GFAP and NSE, markers not expected to be expressed by fibroblasts [17]. Together with the negative PCR result for BPV14, these immunohistochemical findings exclude the diagnosis of feline sarcoid. To avoid overdiagnosis of feline sarcoid in a diagnostic setting, it is therefore strongly advisable to confirm the suspicion of a feline sarcoid using immunohistochemistry and detection of BPV14.

In our study, Sox10 was positive in a total of 25 tumors (73.5%). Sox10 is a protein expressed in neural crest stem cells and is used as a melanocytic marker in humans and dogs, displaying a similar sensitivity as MelanA and PNL2 [18,19,20,21,22]. In human medicine, it has also been used in combination with other markers for the diagnosis of PNSTs [23,19]. In dogs, Sox10 protein has been detected in peripheral nerves and associated tumors and has been proposed as a discriminative marker between canine PNST and perivascular wall tumor (PWT) [24,21]. Sox10 has also been published as a useful marker in the diagnosis of a PNST in the limb of a pig [25]. There is limited information about the expression of Sox10 in feline tumors, but it has also been utilized, in combination with other antibodies, for the diagnosis of nerve cell tumors in cats [17,26,27]. Although Sox10 has not been confirmed as a marker for feline melanocytic tumors, it is expected to be positive as well, raising the doubt of a spindle amelanotic melanoma as a differential diagnosis in our cases [28,29]. Even if peripheral nerve cells and melanocytes have been proven to be linked in development and disease [30], amelanotic melanomas in cats have a poorer prognosis than low- and intermediate-grade PNSTs [31,11], emphasizing the importance of differentiating between both. All Sox10-positive tumors were negative for Melan A (tested in all 25 tumors) and PNL2 (tested in 22 out of these 25 tumors), largely excluding melanocytic differentiation [32,33]. In addition, most of these tumors presented low to moderate pleomorphism and few mitoses (grade 1 or 2 following Dobromylskyj’s grading system [11]), while amelanotic melanomas in cats typically display high pleomorphism and frequent mitoses [28,31,34,35]. In addition, contrary to humans and dogs, feline amelanotic melanomas have been reported to be of the signet ring or balloon cell type, while spindle cell melanomas were melanized [36]. Furthermore, all 25 tumors were positive for at least one of the other antibodies used in the study. Fifteen tumors, including the two grade 3 tumors following Dobromylskyj’s grading system [11], were positive for GFAP, and one of them was also positive for periaxin, both representing nervous markers that would not be expected to be expressed by neoplastic melanocytes [37,38,39,40]. Moreover, six and three tumors were positive for SMA and p63, respectively. Positivity for these markers has been described in PNST in cats [17] and humans [41], respectively, but is unexpected in melanocytic tumors [32,33]. In summary, these 25 Sox10 positive tumors in our study (73.5%) were interpreted as PNSTs.

Melan A and S100 were shown to be good immunohistochemical markers for feline melanomas, but while being highly sensitive, S100 has been proven to be poorly specific for melanocytic tumors [32]. Following the publication by Ramos-Vara in 2002, reported feline amelanotic melanomas have been diagnosed based on positivity for S100 only [31,29]. In the present study, it has been shown that S100-positive spindle cell tumors in cats might not necessarily be of melanocytic origin, but that PNST must be considered an important differential diagnosis. In consequence, further markers for nervous and melanocytic protein expression should be used to reach a final diagnosis. One tumor in our study was only positive for vimentin and NSE, and another one was only positive for vimentin, S100, and SMA. Although neither NSE nor S100 are specific markers for neoplasia of nervous origin, together with the morphological features, these two tumors were mostly compatible with PNST, making a total of 79.4% (27/34) PNSTs in our study. Feline facial PNSTs have been previously described [17,39], but these studies also included other body locations. Interestingly, melanomas and sarcoids have also been described as frequently affecting the face in cats [8,28,29]. 

Follow-up was available for 13 cats with tumors diagnosed as PNST. All died during the study; eight of them were euthanized due to poor life quality caused by the tumor, including difficulty drinking and eating in those cases in which tumors were located near the mouth (with a survival mean and median of 190 and 115 days, respectively), and the others due to other or unknown reasons. Four of these tumors had relapsed, and one of them likely developed liver metastases, although definitive histopathological confirmation was lacking. One tumor in the study was only positive for vimentin and SMA and diagnosed as a leiomyosarcoma [42], although a fibroblastic tumor with myofibroblast reaction could not be discarded [43]. No follow-up information was available for this cat. 

The four cats with multiple clinically independent facial tumors were all Sox10-positive and interpreted as PNSTs. In three of these cats, the tumors differed in reaction to further antibodies. This could be interpreted as the occurrence of multiple, independent PNSTs. Cats affected by more than one PNST have been reported [39], but the concrete location of the tumors was not specified. In humans, multiple PNSTs occur in patients with neurofibromatosis type 1, a genetic tumor predisposition syndrome, but a similar genetic disorder has not been described in cats [44]. Interestingly, we observed whorling of neoplastic cells around the adnexal structures only in the group of PNSTs positive for Sox10. Whorling of neoplastic cells, including whorls around axons, has been previously described in animals with PNST [24,10]. However, as far as we know, whorling around cutaneous adnexal units, as described in our cases, has not been reported before. These periadnexal whorls were observed in five of the six tumors presenting with atypical clinical signs, including erosion, ulceration, or plaque, including the cat with only poorly demarcated lesions (cat 1), in which most of the neoplastic cells were located around multiple adnexal units without the formation of an obvious nodular lesion. Based on these findings, we hypothesize that periadnexal whorling may be the initial presentation of a subgroup of feline facial PNSTs, which may originate from the axons surrounding the hair follicles and their adnexal glands. This may be a useful histopathological feature for the diagnosis of cutaneous PNST, but further studies are warranted.

Six PNSTs in our study presented clinically first as erosion, ulceration, or plaque, further progressing to an obvious mass in four cases. This is a feature relevant to general practitioners, dermatologists, and oncologists that, as far as we know, has not yet been reported. 

This study has highlighted the histomorphologic variability of PNST in cats. Most of the tumors were low-grade soft tissue sarcomas, but some tumors displayed histologic features of malignancy. Due to the retrospective setting of the study, the clinical follow-up information in these cases is limited, which limits the study of the biologic behavior of these tumors. In one tumor, metastases to the liver were suspected based on radiographic imaging. However, this was not confirmed histologically. Necropsies of cats euthanized due to facial PNST would provide a more complete picture of the biologic behavior of these tumors and would permit a more extensive pathologic analysis of the tumor and possible metastases.

## 5. Conclusions

This study emphasizes the importance of distinguishing feline facial STSs, mainly PNSTs, from feline sarcoids by additional immunohistochemical and BPV14 analysis due to the similar clinical and histomorphological presentation but different prognosis. Furthermore, this study shows that PNST is an important but underdiagnosed feline facial spindle cell tumor type that may present as single or multiple masses on the face of middle-aged cats and also as erosive and ulcerative lesions and plaques. Our study also highlights periadnexal whorling of neoplastic cells as a new, additional histopathological finding in feline facial PNSTs and Sox10 as a useful complementary immunohistochemical marker as part of a panel for the diagnosis of facial PNSTs in cats.

## Figures and Tables

**Figure 1 animals-14-01103-f001:**
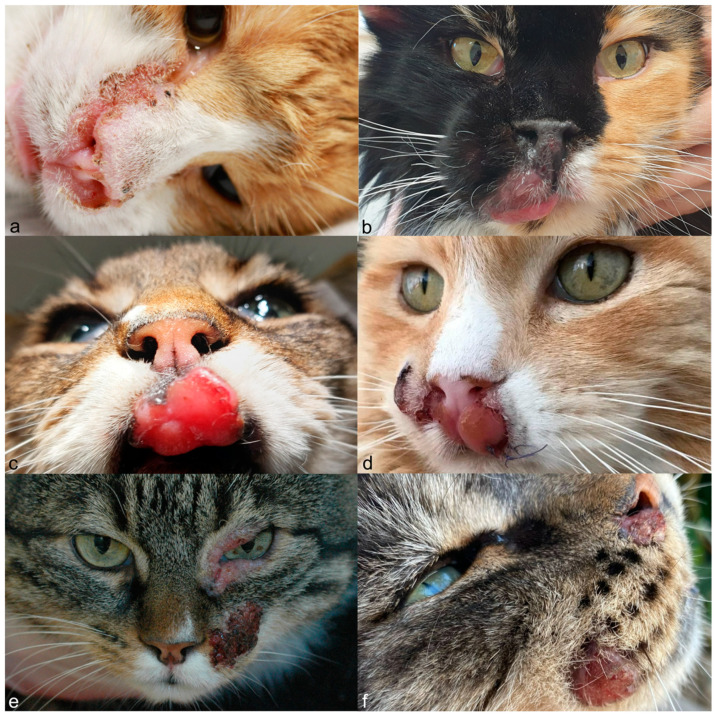
Clinical features of cats with facial spindle cell tumors. Poorly demarcated lesion with crusts, erosions, and ulcerations in cat 1 (**a**). A single mass in cat 2, which has evolved from erythema and erosion (**b**), and cat 8 (**c**). Masses that have evolved from ulcerations and crusts in cat 3 (**d**). A plaque (periocular) and mass (left cheek) in cat 12 (the third tumor in the left upper lip of this cat is not shown) (**e**). Two plaques in cat 20 (**f**). The plaque on the upper lip evolved later into a big mass.

**Figure 2 animals-14-01103-f002:**
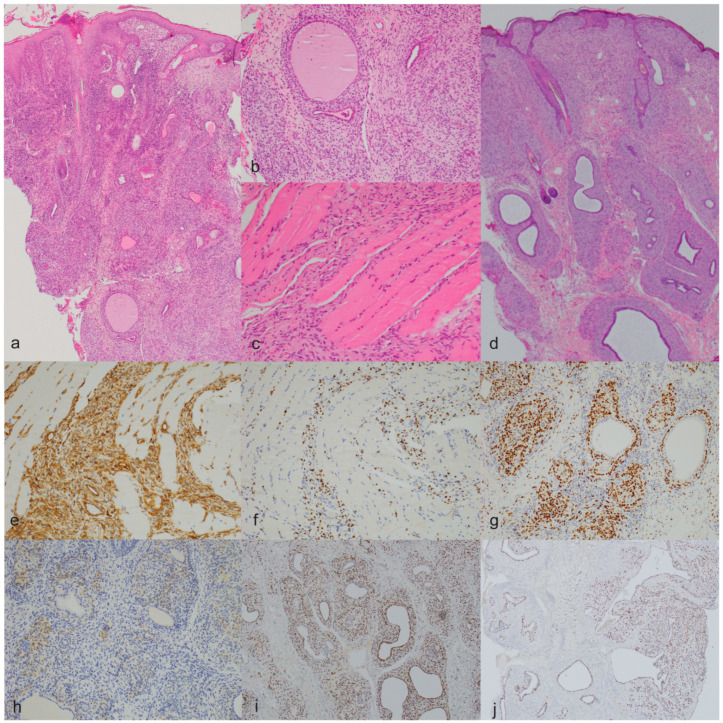
Histomorphological and immunohistochemical features of feline facial spindle cell tumors interpreted as PNSTs with prominent periadnexal whorling. ((**a**), ×40) H&E cat 1, the epidermis is markedly hyperplastic with multifocal crusts. The tumor is characterized by a poorly demarcated proliferation of spindle cells arranged mainly around the adnexal units (periadnexal whorls) ((**b**), ×200) and infiltrating the deeper skeletal muscle ((**c**), ×200); (**d**), ×40) H&E cat 6, also displaying epidermal hyperplasia and neoplastic whorling around adnexal units. (**e**–**h**) Immunohistochemical profile cat 1, the neoplastic cells are positive for vimentin ((**e**), ×200), Sox10 ((**f**,**g**), ×200), and GFAP ((**h**), ×200). This tumor is also positive for S100. (**i**,**j**) Immunohistochemical profile cat 6, the neoplastic cells are positive for Sox 10 ((**i**), ×40) and p63 ((**j**), ×40). This tumor is also positive for vimentin, S100, and GFAP.

**Figure 3 animals-14-01103-f003:**
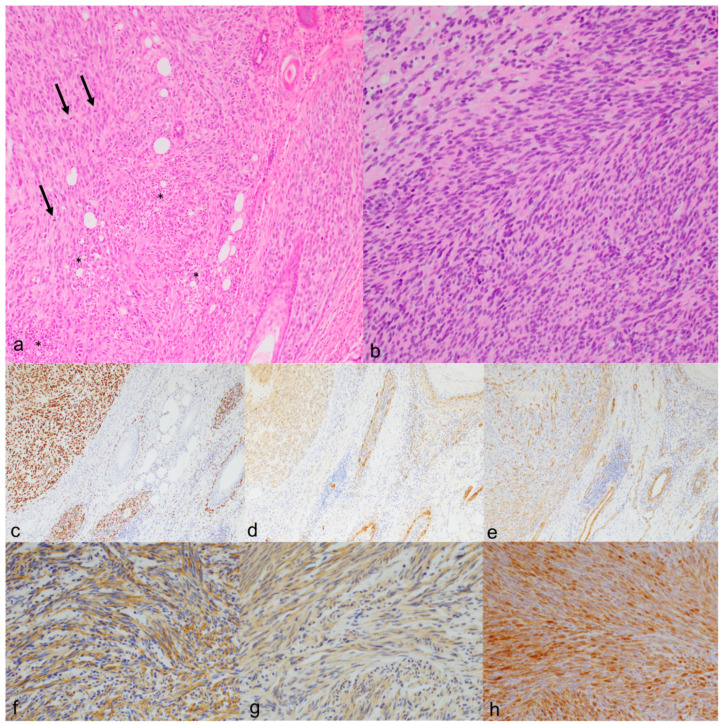
Histomorphological and immunohistochemical features of facial spindle cell tumors grade 1 and 3 interpreted as PNSTs ((**a**), ×100) H&E cat 13, grade 3 PNST. The tumor is partly arranged in sheets (**left**) and partly around adnexal units (**right**). It displays high mitotic activity (arrows) and necrosis (asterisks). ((**b**), ×200) H&E cat 15, grade 1 PNST. The tumor is composed of spindle cells growing in streams with low anisocytosis and anisokaryosis. (**c**–**e**) Immunohistochemical profile, cat 13. The neoplastic cells are positive for Sox10 ((**c**), ×100), S100 ((**d**), ×100), and SMA ((**e**), ×100). This tumor is also positive for vimentin, GFAP, and NSE. f-h) Immunohistochemical profile, cat 15. The neoplastic cells are positive for GFAP ((**f**), ×200), NSE ((**g**), ×200), and periaxin ((**h**), ×200). This tumor is also positive for vimentin, Sox10, and S100.

**Table 1 animals-14-01103-t001:** Primary antibodies and pretreatment protocols used for immunohistochemistry.

Antibody	Species	Clone	Source ^(1)^	Antibody Dilution	Pretreatment Method ^(2)^
Vimentin	Mouse	V9	Dako	1:1000	H2(10)95
Sox10	Mouse	EP268	Cell Marque	1:400	H2(40)95
S100	Mouse	EP32	Cell Marque	1:800	H2(40)95
GFAP	Mouse	6F2	Dako	1:50	H1(20)
NSE	Mouse	MRQ-55	Cell Marque	1:100	H2(20)95
Periaxin	Rabbit	Polyclonal	Sigma-Aldrich	1:200	Pressure cooker, 20 min, 98 °C, Ph9
SMA	Mouse	1A4	Cell Marque	1:500	H2(20)95
p63	Mouse	4A4	Bio SB	1:800	H2(40)95
MelanA	Mouse	A103	Leica Novocastra	1:100	H2(20)95
PNL2	Mouse	PNL2	Cell Marque	1:100	H2(20)95

^(1)^ Company producing the antibody, ^(2)^ H1(20): Pretreatment with Epitope Retrieval Buffer Type 1 (Citrate, pH 6, Leica Biosystems^®^) for 20 min at 100 °C on Bond-III immunostainer (Leica Biosystems^®^), H2(10)95: Pretreatment with Epitope Retrieval Buffer Type 2 (Tris-EDTA, pH 9, Leica Biosystems^®^) for 10 min at 95 °C on Bond-III immunostainer (Leica Biosystems^®^), H2(20)95: Pretreatment with Epitope Retrieval Buffer Type 2 (Tris-EDTA, pH 9, Leica Biosystems^®^) for 20 min at 95 °C on Bond-III immunostainer (Leica Biosystems^®^), H2(40)95: Pretreatment with Epitope Retrieval Buffer Type 2 (Tris-EDTA, pH 9, Leica Biosystems^®^) for 40 min at 95 °C on Bond-III immunostainer (Leica Biosystems^®^).

**Table 2 animals-14-01103-t002:** Clinical information of 29 cats presenting soft tissue tumors (STSs) on the face.

Case	Breed	Sex	Age	Tumors	Location	Clinical Features of the Tumor	Dead/Alive	Cause of Death
1	DSH	mc	14 y	Poorly demarcated lesions	Right medial canthus, nasal bridge, nasal planum, right and left nares	Erosion, ulceration and crusts	Dead	Euthanasia (cardiomyopathy)
2	Maine Coon	fc	3 y	1	Nasal planum and upper lip	Erythema and erosion evolving into mass	Dead	Euthanasia (tumor)
3	Mixed breed	mc	12 y	2	Right nares/left upper lip	Ulceration and crusts evolving into mass (both)	Dead	Euthanasia (tumor)
4	Maine Coon	mc	10 y	1	Lower lip	Mass	Dead	Euthanasia (tumor)
5	Canadian Sphynx	mc	10 y	1	Nasal planum and nasal bridge	Mass	Dead	Unknown
6	DSH	mc	12 y	1	Upper lip	Mass	Dead	Euthanasia (tumor)
7	ESH	fnc	Unknown	1	Forehead left	Mass	Unknown	n/a
8	DSH	mc	13 y	1	Upper lip	Mass	Unknown	n/a
9	ESH	fc	15 y	1	Nasal bridge	Mass	Dead	Euthanasia (tumor)
10	ESH	mc	14 y	2	Left cheek/right cheek	Mass	Dead	Euthanasia (tumor)
11	DSH	mc	14 y	1	Upper lip	Mass	Dead	Unknown
12	DSH	mnc	9 y	3	Left medial canthus extending in upper and inferior eyelids/left upper lip/left cheek	Plaque/Mass/Mass	Dead	Unknown
13	ESH	fc	15 y	1	Lip	Mass	Unknown	n/a
14	ESH	fc	4.5 y	1	Nasal bridge	Mass	Unknown	n/a
15	ESH	mnc	Unknown	1	Lip	Mass	Unknown	n/a
16	DSH	mnc	12 y	1	Lip	Mass	Unknown	n/a
17	BSH	mc	6y	1	Nasal bridge	Mass	Unknown	n/a
18	ESH	fnc	15y	1	Lip	Mass	Unknown	n/a
19	ESH	fc	16 y	1	Lip	Mass	Unknown	n/a
20	Maine Coon	mc	16 y	2	Right nare/right upper lip	Erythematous plaque with evolution into mass of the lip tumor	Dead	Euthanasia (tumor)
21	ESH	Unknown	1 y	1	Left nares	Mass	Alive	n/a
22	ESH	mc	9 y	1	Forehead	Mass	Dead	Euthanasia (tumor)
23	ESH	mc	2 y	1	Upper lip	Mass	Alive	n/a
24	ESH	mnc	6 m	1	Nasal planum	Mass	Dead	Unknown
25	ESH	mc	12 y	1	Nose	Mass	Dead	Euthanasia (abdominal lymphoma)
26	BSH	fc	9 m	1	Left upper lip	Mass	Alive	n/a
27	ESH	mc	1 y	1	Lip	Mass	Unknown	n/a
28	ESH	fc	Unknown	1	Lip	Mass	Unknown	n/a
29	ESH	mnc	2.5 y	1	Lip	Mass	Unknown	n/a

Abbreviations: DSH: Domestic short hair cat; ESH: European short hair cat; BSH: British short hair cat; f: female, m: male, c: castrated; nc: non-castrated; n/a, not applicable.

**Table 3 animals-14-01103-t003:** Histopathological findings and results of the PCR for bovine papillomavirus 14 (BVP14) and immunohistochemical markers for 34 soft tissue tumors (STSs) affecting the face of 29 cats.

Case	Tumor	Histopathological Diagnosis (HE-Based)	Histo Grade ^a^	Whorling ^b^	BPV 14	Vim.	Sox10	S100	GFAP	NSE	Periaxin	MelanA	PNL2	SMA	p63	Final Diagnosis
Cat 1	1	STS	1	Yes	Neg.	Pos.	Pos.	Pos.	Pos.	Neg.	Neg.	Neg.	Neg.	Neg.	Neg.	PNST
Cat 2	2	STS	2	NI	Neg.	Pos.	Pos.	Pos.	Neg.	Neg.	Neg.	Neg.	Neg.	Neg.	Neg.	PNST
Cat 3	3	STS	1	Yes	I	Pos.	Pos.	Pos.	Neg.	Neg.	Neg.	Neg.	Neg.	Neg.	Neg.	PNST
	4	STS	1	Yes	I	Pos.	Pos.	Pos.	Neg.	Neg.	Neg.	Neg.	Neg.	Neg.	Neg.	PNST
Cat 4	5	Feline sarcoid	2	No	I	Pos.	Pos.	Pos.	Pos.	Pos.	Neg.	Neg.	Neg.	Neg.	Neg.	PNST
Cat 5	6	STS vs. AM	3	No	Neg.	Pos.	Pos.	Pos.	Neg.	Neg.	Neg.	Neg.	Neg.	Pos.	Neg.	PNST
Cat 6	7	STS	1	Yes	Neg.	Pos.	Pos.	Pos.	Pos.	Neg.	Neg.	Neg.	Neg.	Neg.	Pos.	PNST
Cat 7	8	Feline sarcoid	1	Yes	Neg.	Pos.	Pos.	Pos.	Pos.	Pos.	Neg.	Neg.	Neg.	Neg.	Neg.	PNST
Cat 8	9	STS	1	NI	Neg.	Pos.	Pos.	Pos.	Neg.	Pos.	Neg.	Neg.	Neg.	Neg.	Neg.	PNST
Cat 9	10	STS	2	NI	Neg.	Pos.	Pos.	Pos.	Pos.	Neg.	Neg.	Neg.	Neg.	Neg.	Pos.	PNST
Cat 10	11	STS	1	Yes	Neg.	Pos.	Pos.	Neg.	Neg.	Neg.	Neg.	Neg.	Neg.	Pos.	Neg.	PNST
	12	STS	1	No	Neg.	Pos.	Pos.	Pos.	Pos.	Neg.	NA	Neg.	Neg.	Neg.	Neg.	PNST
Cat 11	13	Feline sarcoid	2	Yes	Neg.	Pos.	Pos.	Pos.	Pos.	Pos.	Neg.	Neg.	Neg.	Pos.	Pos.	PNST
Cat 12	14	Feline sarcoid	1	No	Neg.	Pos.	Pos.	Pos.	Neg.	Neg.	Neg.	Neg.	Neg.	Neg.	Neg.	PNST
	15	Feline sarcoid	1	No	Neg.	Pos.	Pos.	Pos.	Neg.	Neg.	Neg.	Neg.	Neg.	Neg.	Neg.	PNST
	16	Feline sarcoid	1	Yes	Neg.	Pos.	Pos.	Pos.	Pos.	Neg.	Neg.	Neg.	Neg.	Pos.	Neg.	PNST
Cat 13	17	STS vs. AM	3	Yes	Neg.	Pos.	Pos.	Pos.	Pos.	Pos.	Neg.	Neg.	Neg.	Pos.	Neg.	PNST
Cat 14	18	STS	1	No	Neg.	Pos.	Pos.	Pos.	Pos.	Neg.	Neg.	Neg.	Neg.	Neg.	Neg.	PNST
Cat 15	19	STS	1	NI	Neg.	Pos.	Pos.	Pos.	Pos.	Pos.	Pos.	Neg.	NA	Neg.	Neg.	PNST
Cat 16	20	STS	1	NI	Neg.	Pos.	Pos.	Pos.	Pos.	Neg.	Neg.	Neg.	Neg.	Neg.	Neg.	PNST
Cat 17	21	STS	1	No	Neg.	Pos.	Pos.	Pos.	Pos.	Neg.	Neg.	Neg.	Neg.	Pos.	Neg.	PNST
Cat 18	22	STS	2	No	Neg.	Pos.	Pos.	Pos.	Pos.	Neg.	Neg.	Neg.	NA	Neg.	Neg.	PNST
Cat 19	23	STS	1	No	Neg.	Pos.	Pos.	Pos.	Neg.	Neg.	Neg.	Neg.	NA	Neg.	Neg.	PNST
Cat 20	24	STS	1	Yes	Neg.	Pos.	Pos.	Pos.	Pos.	Neg.	NA	Neg.	Neg.	Neg.	Neg.	PNST
	25	STS	1	Yes	Neg.	Pos.	Pos.	Pos.	Neg.	Neg.	NA	Neg.	Neg.	Neg.	Neg.	PNST
Cat 21	26	Feline sarcoid	1	No	Pos.	Pos.	Neg.	Neg.	Neg.	Neg.	NA	Neg.	NA	Neg.	Neg.	Feline sarcoid
Cat 22	27	STS	1	No	Neg.	Pos.	Neg.	Neg.	Neg.	Pos.	Neg.	Neg.	Neg.	Neg.	Neg.	PNST
Cat 23	28	Feline sarcoid	1	NI	Pos.	Pos.	Neg.	Neg.	Neg.	Neg.	Neg.	Neg.	NA	Neg.	Neg.	Feline sarcoid
Cat 24	29	Feline sarcoid	1	NI	Pos.	Pos.	Neg.	Neg.	Neg.	Neg.	Neg.	Neg.	NA	Neg.	Neg.	Feline sarcoid
Cat 25	30	STS	1	NI	Neg.	Pos.	Neg.	Pos.	Neg.	Neg.	Neg.	Neg.	Neg.	Pos.	Neg.	PNST
Cat 26	31	Feline sarcoid	1	No	Neg.	Pos.	Neg.	Neg.	Neg.	Neg.	Neg.	Neg.	NA	Neg.	Neg.	Feline-sarcoid like
Cat 27	32	Feline sarcoid	1	No	Pos.	Pos.	Neg.	Neg.	Neg.	Neg.	Neg.	Neg.	NA	Neg.	Neg.	Feline sarcoid
Cat 28	33	STS	1	NI	Neg.	Pos.	Neg.	Neg.	Neg.	Neg.	Neg.	Neg.	Neg.	Pos.	Neg.	Leiomyosarcoma
Cat 29	34	Feline sarcoid	1	No	Pos.	Pos.	Neg.	Neg.	Neg.	Neg.	Neg.	Neg.	NA	Neg.	Neg.	Feline sarcoid
Total					5/31	34/34	25/34	25/34	15/34	7/34	1/30	0/34	0/25	8/34	3/34	

Abbreviations: ^a^, histopathological grading according to the grading system for cutaneous and subcutaneous soft tissue sarcomas proposed by Dobromylskyj et al., 2021 [11]; ^b^, periadnexal whorling; HE, hematoxylin and eosin stain; BVP14, PCR for bovine papillomavirus 14; Vim., vimentin; Neg., negative; Pos., positive; STS, soft tissue tumor/soft tissue sarcoma; AM, amelanotic melanoma; I, inconclusive; NI, not interpretable; NA, not available.

## Data Availability

The data presented in this study are available on request from the corresponding author.

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
