# Peer review of "Feline Facial Spindle Cell Tumors in 29 Cats: Histomorphological and Immunohistochemical Characterization"

_animals, 2024, doi:10.3390/ani14071103_

Round 1

Reviewer 1 Report

Comments and Suggestions for Authors

This is a well written paper. Table 2 and the figures are especially helpful and well done. While it is not the purpose of this paper, you could consider adding in time from diagnosis to euthanasia in cases where you have that information.

Line 138Please elaborate on the known positive control

Line 363 Please add another reference for "Even if peripheral nerve cells and melanocytes have been proved to be linked in development and disease, amelanotic melanomas have a poorer prognosis than PNSTs, emphasising the importance of differentiating between both." The current reference (28) does not address that amelanotic melanomas have a poorer px than PNST (may also need to specify the species that you are referring to)

Comments on the Quality of English Language

Minor edits needed

Author Response

Comments and Suggestions for Authors

Thank you very much for your feedback. As requested, please find our answers point-by-point in blue.

This is a well written paper. Table 2 and the figures are especially helpful and well done. While it is not the purpose of this paper, you could consider adding in time from diagnosis to euthanasia in cases where you have that information. The survival time from diagnosis to euthanasia is included in the follow-up results. For clarity, we specified it more clearly in this section of the manuscript (actual lines 340-341).

-Line 138Please elaborate on the known positive control. The immunohistochemical tissue controls have been specified in the material and methods section (actual lines 148-151).

-Line 363 Please add another reference for "Even if peripheral nerve cells and melanocytes have been proved to be linked in development and disease, amelanotic melanomas have a poorer prognosis than PNSTs, emphasising the importance of differentiating between both." The current reference (28) does not address that amelanotic melanomas have a poorer px than PNST (may also need to specify the species that you are referring to). This reference is valid for the first part of the sentence. For the second part, another two references have been included (actual lines 405-406).

Comments on the Quality of English Language

Minor edits needed

Reviewer 2 Report

Comments and Suggestions for Authors

The present study addresses a very interesting topic in veterinary oncology. I made some comments below hoping they can be helpful for the authors. As a general comment, please revise the journals’ Instructions for Authors to amend in-text citation style.

Line 2. The title could be improved by clearly mentioning what was the aim of the study or the main findings. For example, a suggestion could be "Immunohistochemical and histomorphological characterization of feline facial spindle cell tumors”.

Line 19 – 20. In addition to my previous observation, the current aim of the study seems too general. It could also be improved to reflect that the authors are making an immunohistochemical and histomorphological characterization of soft tissue tumors in cats. Consider making the suggested change and implementing it in all places where the aim is mentioned.

Lines 20 – 21. I consider that this introductory sentence is not necessary for this section. The authors could directly start by mentioning the number of animals or tumors that were evaluated in the study. Additionally, the authors could specify some demographic characteristics of the animals (e.g., age, sex, and breed)

Line 46. I suggest including the keyword "oncology" to improve the recognition of the present study in search databases.

Line 66. To improve the connection between paragraphs, I recommend mentioning the incidence and impact on the animals' health when dealing with this type of tumor. Another issue that is important to mention is the lack of studies describing the characteristics of the tumor and how this could influence the treatment or therapeutic management.

Lines 74- 76. Please, consider improving the objective of the study as mentioned above.

Line 79. I suggest including the demographic characteristics of the study animals. Also, add some general information from where these patients were obtained (e.g., clinics, hospitals. Information regarding the tumor such as time of diagnosis to beginning of the study, if the cats were under treatment, if the tumors were obtained from live animals or during necropsy, etc.

Line 312- 315. Consider deleting these lines and start by mentioning that “Based on PCR results…”

Line 323. When mentioning that five cats up to 2.5 years were confirmed with feline sarcoid, it would be interesting if the authors could discuss why this type of tumor is present in relatively young cats by providing a possible biological explanation.

Lines 397 – 404. Similar to my previous comment, a discussion on the pathophysiology of these types of tumors and why high mortality rates are observed would improve the discussion.

Line 428. Before the conclusion, I suggest adding some limitations and possible perspectives or issues that need further investigation according to the present findings.

Author Response

Comments and Suggestions for Authors

Thank you very much for your feedback. As requested, please find our answers point-by-point in blue.

The present study addresses a very interesting topic in veterinary oncology. I made some comments below hoping they can be helpful for the authors. As a general comment, please revise the journals’ Instructions for Authors to amend in-text citation style. We used Mendeley and a free style, as recommended in the journal instructions. We are sorry that we had not properly checked that Mendeley had good formatting for all references. We checked them and corrected the ones that had an improper format.

-Line 2. The title could be improved by clearly mentioning what was the aim of the study or the main findings. For example, a suggestion could be "Immunohistochemical and histomorphological characterization of feline facial spindle cell tumors”.  We changed the title to “Feline facial spindle cell tumors in 29 cats: Histomorphological and immunohistochemical characterization”

-Line 19 – 20. In addition to my previous observation, the current aim of the study seems too general. It could also be improved to reflect that the authors are making an immunohistochemical and histomorphological characterization of soft tissue tumors in cats. Consider making the suggested change and implementing it in all places where the aim is mentioned. We changed it as suggested in the lines 19-20 (actual lines 20-21) and in the lines 74-76 (actual lines 75-78). We consider that the aim was already described as suggested in the rest of the manuscript.

-Lines 20 – 21. I consider that this introductory sentence is not necessary for this section. The authors could directly start by mentioning the number of animals or tumors that were evaluated in the study. Additionally, the authors could specify some demographic characteristics of the animals (e.g., age, sex, and breed). The mentioned introductory sentence has been deleted. As the number of words for the simple and extended abstracts is limited and as the findings in the demographic characteristic of the animals were not considered major, we concluded to better not specify demographic characteristics here.

-Line 46. I suggest including the keyword "oncology" to improve the recognition of the present study in search databases. “Oncology” has been included as keyword.

-Line 66. To improve the connection between paragraphs, I recommend mentioning the incidence and impact on the animals' health when dealing with this type of tumor. Another issue that is important to mention is the lack of studies describing the characteristics of the tumor and how this could influence the treatment or therapeutic management. We rephrase this line and in accordance also the introduction to the previously described STSs (actual lines 68-69, 52). We favored not to pronounce about the lack of studies about this tumor, as we consider that there are some publications about this neoplasia.

-Lines 74- 76. Please, consider improving the objective of the study as mentioned above. As mentioned in the previous comment, it has been modified (actual lines 75-78).   

-Line 79. I suggest including the demographic characteristics of the study animals. Also, add some general information from where these patients were obtained (e.g., clinics, hospitals). Information regarding the tumor such as time of diagnosis to beginning of the study, if the cats were under treatment, if the tumors were obtained from live animals or during necropsy, etc. We consider that the demographic characteristics of the animals do not belong to the Material&Methods, as suggested by the reviewer, but to the results section, where they are already described, both in the text and in table 2. A sentence indicating that we investigated biopsy samples and about the origin of the samples has been included at the beginning of the section. We have also included the year in which the biopsies were sampled (actual lines 83-90).. The scant information about treatment we could get is already included in the follow-up results. Not more information about treatment is available.

-Line 312- 315. Consider deleting these lines and start by mentioning that “Based on PCR results…” In case the number of words in the manuscript is not a problem for the journal, we would prefer to maintain this introductory sentence in the discussion section.

-Line 323. When mentioning that five cats up to 2.5 years were confirmed with feline sarcoid, it would be interesting if the authors could discuss why this type of tumor is present in relatively young cats by providing a possible biological explanation. It has been included (actual lines 365-366)

-Lines 397 – 404. Similar to my previous comment, a discussion on the pathophysiology of these types of tumors and why high mortality rates are observed would improve the discussion. Unfortunately, our findings of the retrospectively assessed clinical follow-up were not extensive enough to discuss about the pathophysiology of these tumors. We added this as a limitation of the study in the last paragraph of the discussion. In the manuscript it was included that the cause of euthanasia is the poor-quality life. A short sentence extending this has been added (actual lines 439-440).

-Line 428. Before the conclusion, I suggest adding some limitations and possible perspectives or issues that need further investigation according to the present findings. Limitations have been included (actual lines 469-477).

Reviewer 3 Report

Comments and Suggestions for Authors

Dear Authors,
thank you for this well-written overview on feline facial spindle cell tumors. The English language used is appropriate, there are only a few typos I will detail. Also, please take out the abbreviations from the Simple abstract as this is the journal's requirement.

- line 194: "eight (23.5%) not specified); nine (26.5%) at the nose (three (8.8%) on the nasal bridge..." - no need of the highlighted brackets

- line 197: "specified); one (2.9%) at the nasal planum and upper lip..." - no need of the highlighted bracket

- line 204: "and neoplastic (squamous 203 cell carcinoma, feline sarcoid, not further specified). - highlighted bracket is missing at the end of sentence.

- Material and methods: please specify for every PCR, how much base pairs are the expected amplicons.

- Figures 2 and 3: please specify the magnification for every histopathologic image and which tools were used to digitalize the slides and take the pictures (can be added to the related chapter of Material and methods).

- Table 1: it would be beneficial to add the information where to expect specific staining with the antibodies used in the study.

- Figure 3/a: necrotic areas can hardly be seen, I would suggest a higher magnification image.

Overall this is a very interesting manuscript, the study gives useful information to clinicians, oncologists and pathologists.

Author Response

Comments and Suggestions for Authors

Thank you very much for your feedback. As requested, please find our answers point-by-point in blue.

Dear Authors,

thank you for this well-written overview on feline facial spindle cell tumors. The English language used is appropriate, there are only a few typos I will detail. Also, please take out the abbreviations from the Simple abstract as this is the journal's requirement.

- line 194: "eight (23.5%) not specified); nine (26.5%) at the nose (three (8.8%) on the nasal bridge..." - no need of the highlighted brackets It has been corrected (actual line 210).

- line 197: "specified); one (2.9%) at the nasal planum and upper lip..." - no need of the highlighted bracket. It has been corrected (actual lines 163 and 171-172).

- line 204: "and neoplastic (squamous 203 cell carcinoma, feline sarcoid, not further specified). - highlighted bracket is missing at the end of sentence. It has been corrected (actual line 220).

- Material and methods: please specify for every PCR, how much base pairs are the expected amplicons. This information has been added (actual lines   ). In relation with it, we have added in the PCR results a sentence regarding the identity to the published BPV14 (actual lines 163 and 171-172). The person who helped us with this information has been included in the acknowledgment.

- Figures 2 and 3: please specify the magnification for every histopathologic image and which tools were used to digitalize the slides and take the pictures (can be added to the related chapter of Material and methods). The magnification of the images has been included in the figure legends. The tools used to take the microscopic pictures have been detailed in the material and methods section (actual lines 99-100, 156-157).

- Table 1: it would be beneficial to add the information where to expect specific staining with the antibodies used in the study. The cellular localization of the antibody epitopes has been included in the section “Pathological findings” (actual lines 294-295).

- Figure 3/a: necrotic areas can hardly be seen, I would suggest a higher magnification image. We are still of the opinion that the necrotic areas can be observed in the actual figure 3a, but as suggested by the reviewer, a new version of this figure including an inset with a higher magnification of the necrosis has been included. In order to avoid covering important parts of the figure, the inset had to be included on the top of the figure, which is probably not ideal. In consequence, we would favor the first version of the figure, but for us it is also fine if the new version of the figure with the inset is used in the paper.

Overall this is a very interesting manuscript, the study gives useful information to clinicians, oncologists and pathologists.

Round 2

Reviewer 2 Report

Comments and Suggestions for Authors

I believe the authors were very diligent in modifying the paper based on my comments. Therefore, I think the paper should be published. I have no additional comments. Regards.